# Arterial Elasticity in Ehlers-Danlos Syndromes

**DOI:** 10.3390/genes11010055

**Published:** 2020-01-04

**Authors:** Amanda J. Miller, Jane R. Schubart, Timothy Sheehan, Rebecca Bascom, Clair A. Francomano

**Affiliations:** 1Department of Neural and Behavioral Sciences, Penn State College of Medicine, Hershey, PA 17033, USA; 2Department of Surgery, Penn State College of Medicine, Hershey, PA 17033, USA; jschubart@pennstatehealth.psu.edu; 3Department of Neurology, Medical University of South Carolina, Charleston, SC 29425, USA; sheehant@musc.edu; 4Department of Medicine, Penn State College of Medicine, Hershey, PA 17033, USA; rbascom@pennstatehealth.psu.edu; 5Department of Medical and Molecular Genetics, Indiana University School of Medicine, Indianapolis, IN 46202, USA; cfrancom@iu.edu

**Keywords:** Ehlers-Danlos syndromes, pulse wave velocity, blood pressure, orthostatic intolerance

## Abstract

Ehlers-Danlos Syndromes (EDS) are a group of heritable disorders of connective tissue (HDCT) characterized by joint hypermobility, skin hyperextensibility, and tissue fragility. Orthostatic intolerance (OI) is highly prevalent in EDS however mechanisms linking OI to EDS remain poorly understood. We hypothesize that impaired blood pressure (BP) and heart rate control is associated with lower arterial stiffness in people with EDS. Orthostatic vital signs and arterial stiffness were assessed in a cohort of 60 people with EDS (49 female, 36 ± 16 years). Arterial elasticity was assessed by central and peripheral pulse wave velocity (PWV). Central PWV was lower in people with EDS compared to reference values in healthy subjects. In participants with EDS, central PWV was correlated to supine systolic BP (r = 0.387, *p* = 0.002), supine diastolic BP (r = 0.400, *p* = 0.002), and seated systolic BP (r = 0.399, *p* = 0.002). There were no significant correlations between PWV and changes in BP or heart rate with standing (*p* > 0.05). Between EDS types, there were no differences in supine hemodynamics or PWV measures (*p* > 0.05). These data demonstrate that increased arterial elasticity is associated with lower BP in people with EDS which may contribute to orthostatic symptoms and potentially provides a quantitative clinical measure for future genotype-phenotype investigations.

## 1. Introduction

Ehlers-Danlos syndromes (EDS) are a collection of heritable disorders of connective tissue characterized by joint hypermobility, mild skin hyperextensibility, and tissue fragility [1]. Common symptoms of EDS include joint instability, chronic pain, gastrointestinal issues, and sleep disturbances [2]. Many people with EDS have persistent symptoms of orthostatic intolerance (OI) including lightheadedness, fatigue, nausea, and palpitations [3]. Additionally, the prevalence of EDS is higher in patients with orthostatic intolerance compared to the general population [4]. The association between EDS and autonomic cardiovascular dysfunction is most prevalent in people with hEDS [5,6,7], but there is also evidence of orthostatic intolerance in classical EDS [8]. The high prevalence of OI in EDS demonstrates a need to understand cardiovascular pathophysiology in all EDS types, as the pathophysiology explaining the high rate of OI in EDS is unknown. The leading theory connecting the two disorders is that generalized connective tissue laxity in EDS increases vascular compliance, leading to insufficient vasoconstriction and venous insufficiency when upright resulting in symptoms of OI [3]. Despite its wide acceptance, there is only data to support this theory in small samples of people with vascular EDS and there is no published evidence to support this theory across other types of EDS.

Pulse wave velocity (PWV) has emerged as the gold standard method for measuring stiffness of the arteries because of its reliability and reproducibility [9]. PWV is a non-invasive technique that involves placing pressure transducers on the skin that can sense the velocity of blood traveling in the arteries, which is a function of the stiffness or elasticity of the arteries. Central PWV, the most widely used and accepted measurement for PWV, measures the stiffness or elasticity of the central cardiovascular system from the carotid to femoral arteries. Using this technique, increased PWV (implying increased arterial stiffness) has been shown to predict future hypertension, coronary heart disease, stroke, adverse cardiovascular events, and mortality [10].

While PWV is well accepted as a measure of arterial stiffness, it has been used far less often to measure arterial elasticity, which is the mathematical inverse of stiffness. More distensible arteries will stretch more as pulse waves travel, resulting in lower (slower) pulse wave velocity. Few studies have sought to identify people, including those with EDS, with suspected increased arterial elasticity and hence decreased PWV. One study evaluated PWV in nine people with comorbid hypermobile EDS and postural tachycardia syndrome (POTS) and found PWV measurements were not different compared to healthy controls [11]. Two studies examined PWV in people with vascular EDS. One study found decreased PWV in about 20% of genetically related people with vascular EDS [12]. The other study found that PWV measurements in people with vascular EDS were similar to those of healthy volunteers [13].

Therefore, the current study is the first assessment of PWV measurements in a large heterogeneous sample of people with EDS. We hypothesized that the collagen changes in EDS would confer an increased distensibility of the vasculature in all EDS types, and that this would contribute to orthostatic intolerance. In this study, we investigated central and peripheral arterial stiffness in people with EDS using the non-invasive measurement of pulse-wave velocity (PWV). We hypothesize that impaired blood pressure (BP) and heart rate control is associated with increased arterial elasticity in people with EDS.

## 2. Materials and Methods

The National Institute on Aging (NIA) study Clinical and Molecular Manifestations of HDCT was designed to investigate the natural history of the most common HDCT. Emphasis was placed on the cardiovascular, musculoskeletal, and neurological complications of HDCT and the natural history of these complications. The original study protocol was designed to collect clinical and family history data, and to use this information to clarify the clinical distinctions between diagnoses. Consenting participants were initially classified based on diagnostic criteria in place at the time of their clinical visit at the NIA (2001–2013). Subjects contributing only biological samples were diagnosed either through a limited onsite evaluation or through review of submitted medical records. The HDCT NIA Dataset v. 2016 consented cohort includes 1009 participants with an average age of 39 ± 18 years (range 2–95, median 40). One hundred and ninety-four participants were 18 years or younger.

The NIA study Clinical and Molecular Manifestations of HDCT began by assembling consented cohorts with a wide range of heritable HDCT, under an umbrella protocol (Protocol 2003-086, later changed to 03-AG-N330). After the study was closed to enrollment, the Institutional Review Board approved the reorganization and migration of the data into a relational database repository and approved re-contacting participants to determine if they would be interested in participating in future research. The HDCT cohort data are provided in SAS datasets, PDF, Excel, MRI DICOM file formats and are now under the umbrella of protocol 11-AG-N079, Sample and Data Repository Protocol for NIA Studies. Participants were recruited from the pool of patients previously seen by the principal investigators and from patient support groups nationally. An authorized guardian provided consent for minor participants, with age-appropriate assent by the minor. In 2016, a signed Data Transfer Agreement between NIA and Penn State University resulted in transfer of a copy of the HDCT NIA Dataset v.2016 data repository to the Penn State University Clinical Translational Science Institute (PSU-CTSI). Datasets were accompanied by copies of original CRFs and SAS dataset codebook descriptions [14].

Participants were stratified by EDS type including: classical, hypermobile, vascular, or other or unclassified according to the Villefranche nosology [15] as previously described for this cohort [14]. Briefly, classical EDS was determined by joint laxity and skin that is extremely hyperextensible, fragile, bruises easily, and has thin atrophic scars. Hypermobile EDS was classified by history of dislocations, generalized joint laxity, and velvety texture of skin with an absence of extreme skin extensibility and profoundly abnormal scars. Vascular EDS was determined by genetic testing for variation in the COL3A1, the gene encoding type III collagen. The other and unclassified EDS category included patients with the rarer types of Ehlers–Danlos syndromes. A molecular diagnosis was used for the arthrochalasia and kyphoscoliotic types. Some patients had features overlapping with two or more types of EDS, and classification proved to be difficult in those cases, and such patients were diagnosed as “EDS, unclassified”. If there was the clinical impression of EDS but they did not meet the diagnostic criteria for any of the known types, we assigned a diagnosis of “EDS, unclassified”.

The analytic cohort for the present study was a subset of the EDS cohort from this NIA study of HDCT consisting of 60 participants who had both orthostatic BP recordings and PWV measurements. BP and heart rate were measured by brachial artery oscillometry in triplicate following 5 min in the supine, seated, and standing postures. Central arterial stiffness was measured by carotid to femoral PWV and peripheral stiffness by carotid to radial PWV.

### 2.1. PWV Measurements

The methods used to assess PWV in this study were the same methods used in the Baltimore Longitudinal Study of Aging [16]. In short, PWV data were collected using a SphygmoCor device (AtCor Medical) that utilizes an EKG and high-fidelity tonometer to acquire waveforms from carotid, femoral, and radial pulses. The software determines the velocity of the pulse wave, i.e., estimated time that it takes the pulse wave to travel between pulse sites divided by the distance between sites. Central PWV is calculated by measuring pulse waves at the carotid and femoral arteries, representing the stiffness of the central vascular tree. Peripheral PWV is calculated by measuring pulse waves at the carotid and radial arteries indicating blood flow to peripheral vascular beds. Reference values of pulse wave velocity in healthy humans were collected using similar methods (pulse wave tonometry divided by distance between sites) [16].

### 2.2. Orthostatic Vital Sign Measurements

Orthostatic vital signs were measured by a brachial artery BP cuff on both arms. BP was measured supine then seated then during standing. Study participants stayed in each posture (supine, sitting, and standing) for 5 min prior to BP recordings. BP was measured in triplicate in each position with one minute between recordings. If BP varied by 15 mmHg or heart rate by 10 beats/minute in one position, a fourth recording was measured. All BP and heart rate measurements on the left arm were averaged for each participant in each posture.

### 2.3. Data Analysis

Descriptive statistics include demographic data and EDS type. Comparison of characteristics among types was performed using ANOVA with post-hoc Tukey-Kramer tests when justified. Pearson’s correlations were run between BP, heart rate, and PWV measurements for the entire cohort. We performed a stratified analysis of central PWV measurements by age in the EDS participants of all types and compared those values to age-matched reference values from a large cohort of healthy participants (*n* = 1455, Reference Values for Arterial Stiffness, 2010) [17].

## 3. Results

Overall our data set included 60 (49 female) EDS participants age 13–70 years. There were no differences in age, height, weight, and body mass index between EDS participants of different types (Table 1).

### 3.1. Pulse Wave Velocity in EDS

Arterial elasticity did not differ by EDS type (Table 1). Grouped together, central PWV is lower in participants with EDS (4.73 ± 0.16 cm/s) compared to reference values in a large sample of healthy participants (Figure 1). PWV increases with age in healthy populations but the increase in arterial stiffness with aging is attenuated in people with EDS.

### 3.2. Orthostatic Blood Pressure in EDS

In the supine posture, BP and heart rate did not vary by EDS type (Table 1). In the standing posture, there was more variability in BP and heart rate measurements within each EDS type as shown by higher standard deviations compared to supine measurements demonstrating a wide range in responses to orthostasis (Table 1). Systolic BP in the standing posture was different between EDS types (ANOVA, *p* = 0.003). Post-hoc analysis showed that standing systolic BP was lower in participants with vascular EDS compared to those with hypermobile EDS (*p* = 0.021) and other/unspecified EDS (*p* = 0.002). Standing diastolic BP and heart rate also trended lower in the vascular EDS group (*p* = 0.087, Table 1.)

### 3.3. Correlations between Pulse Wave Velocity and Blood Pressure

Correlations of central and peripheral PWV to BP and heart rate are shown in Table 2. Central PWV did not correlate to HR or orthostatic BP changes over a 5 min period. Central PWV correlated significantly with supine (*r* = 0.387) and seated (*r* = 0.399) systolic BPs and supine diastolic BP (*r* = 0.400). Peripheral PWV did not correlate to HR or orthostatic BP. Peripheral PWV was correlated to diastolic BP in the supine (*r* = 0.322), seated (*r* = 0.383), and standing (*r* = 0.323) postures. All significant correlations were positive indicating that lower PWV (more elasticity) is associated with lower BP in our cohort of EDS participants.

## 4. Discussion

### 4.1. Overall Findings

This study used PWV to evaluate arterial stiffness in a diverse sample of people with different EDS types. This study provides three novel findings. We demonstrated that PWV is lower in people with EDS compared to reference values in the healthy population implying that their arteries are more elastic. We also found that lower PWV (indicating greater elasticity) is associated with lower systolic and diastolic BP in people with EDS. These findings may help explain the connection between EDS and impaired autonomic cardiovascular control. We also found no differences in PWV measurements among EDS types which suggests that the elasticity of the vasculature is similar among the diverse types of EDS.

### 4.2. Significance of Decreased Pulse Wave Velocity in Ehlers-Danlos Syndrome

The clinical association between EDS and orthostatic intolerance was identified in 1999 by Rowe et al. who first hypothesized that the mechanism connecting these two disorders is an increased enhanced elasticity in the arteries of people with EDS, predisposing them to OI [8]. Two decades later, this theory has become widely accepted despite the lack of empirical data to support it [3].

PWV has become the gold standard for assessing arterial structure because it is reproducible and aligns with more invasive measures. Its ease of use means it is available for testing in larger cohorts [9]. PWV has become a popular and validated method to assess increased stiffness of the central and peripheral vascular system in healthy humans and disease populations ranging from cardiovascular to neurological disease [9,10]. However, this technique is less commonly used to assess populations with increased arterial elasticity.

Three studies have previously measured PWV in people with EDS [11,12,13]. In a single family of 27 people with vascular EDS, Francois et al. utilized an older method for measuring pulse wave velocity involving piezo crystal microphones over the carotid, femoral, and dorsal arteries, and reported significantly decreased PWV (outside 2 standard deviations of normal values) in 5/27 participants studied [12]. A more recent study used the ultrafast ultrasound technique in 102 healthy participants and 37 vascular EDS participants and found that that central PWV was not significantly different in vascular EDS participants compared to controls [13]. Cheng, et al. employed a similar tonometry technique as was used in the current study to assess PWV in nine people with comorbid hypermobile EDS and POTS and nine age, sex, and BMI matched healthy controls, and found a trend to lower central PWV measurements in the people with EDS/POTS compared to controls [11]. Our study adds to the current literature by measuring both central and peripheral PWV in a larger and more heterogeneous group of people with EDS. In contrast to prior studies, we found that central PWV was significantly decreased in people with EDS compared to reference ranges for healthy subjects. This is likely due to our larger and more diverse sample.

The issue of age-associated changes in vascular function in people with vascular EDS was addressed by Mirault et al. using ultrafast ultrasound imaging (a method used by this group to measure PWV). They reported that the age-associated increase in vascular stiffness was attenuated in the vascular EDS participants [13]. We observed a similar phenomenon, namely that the PWV increased very little with progressive age deciles (Figure 1) which differs from reference values derived from healthy humans. High PWV (implying increased arterial stiffness) is related to adverse cardiovascular events in large epidemiology studies [18,19,20]. While one may speculate that lower PWV may be cardio-protective, it is unclear whether increased arterial elasticity is beneficial in people with EDS. Whether PWV has prognostic value in EDS deserves further investigation.

### 4.3. Association between Pulse Wave Velocity and Blood Pressure

Overall, lower PWV is related to lower BP measurements but is not directly indicative of orthostatic tolerance in EDS. These findings are consistent with measurements in healthy subjects and in other patient populations in which PWV tracks similarly to BP [10]. All significant correlations were positive indicating that lower PWV (more elasticity) is associated with lower BP in our cohort of people with EDS. We cannot infer causation from these data.

### 4.4. Comparisons between EDS Types

We did not see a difference in most orthostatic vital signs between EDS types. Systolic BP was slightly lower in vascular EDS which may reflect a difference in physiology or medications taken. Overall, there was a huge range in BP and heart rate responses to orthostasis which demonstrates inconsistent hemodynamic responses in this population and may reflect the presence of different types of OI. According to Roma, et al., about half of people with EDS have POTS (increase in heart rate of 30 beats/minute while standing) but others have orthostatic hypotension or hypertension [3]. It has been thought that vascular EDS was unique in terms of increased arterial distensibility. Our data are the first to compare arterial elasticity among EDS types in a single study, and demonstrate no difference in PWV among types of EDS. This is an important point, and it provides a possible explanation for the common presence of orthostatic intolerance in all EDS types.

### 4.5. Strengths

Strengths of this this study are the large sample size with a diverse EDS cohort including several EDS types, and the concomitant measurement of orthostatic vital signs and PWV. We compared central PWV measurements to published reference values in a large cohort. To our knowledge, there are no peripheral PWV reference values from large populations. The methods used in this study were consistent with study protocols used in the Baltimore Longitudinal Study of Aging [16].

### 4.6. Limitations

This study had several limitations. First, we did not include a contemporaneous control group in this study. However, we used reference values from a large cohort of healthy volunteers for comparison [16]. Second, participants were accessed while on medications which may impact BP, heart rate, and PWV assessments. Orthostatic vital signs were measured following 5 min in the supine posture then after 5 min sitting then 5 min of standing. This limits the ability to diagnose orthostatic intolerance as current diagnostic criteria for orthostatic intolerance involves hemodynamic measurements from the supine to standing posture after at least 10 min [21]. Finally, we acknowledge the heterogeneity of our EDS participants as a potential problem. Since these data were collected prior to the 2017 reclassification of EDS [1,14,15]. It is possible that some participants classified as having hypermobile or unspecified EDS would be categorized as hypermobility spectrum disorders using current criteria.

## 5. Conclusions

Overall, this is the first report of increased arterial elasticity in all types of EDS. The increased arterial elasticity was associated with lower supine and seated systolic and diastolic blood pressure in all types of EDS. We did not see differences in PWV in different types of EDS but standing systolic and diastolic blood pressure were lower in vascular EDS compared to the hypermobile and unspecified types. Our findings suggest that increased arterial elasticity may be related to impaired blood pressure control in EDS. Further studies are needed to determine whether this pathophysiological finding relates to orthostatic symptoms in people with EDS.

## Figures and Tables

**Figure 1 genes-11-00055-f001:**
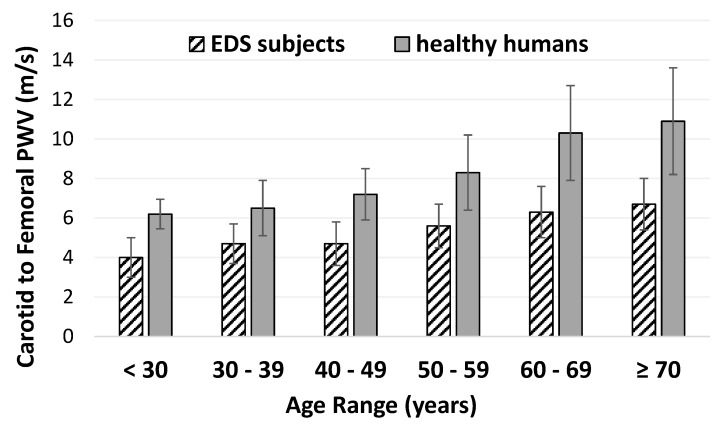
Pulse wave velocity (PWV) by age in participants with Ehlers-Danlos syndromes (EDS) compared to reference values in healthy humans’ data from normal subjects in Reference Values for arterial Stiffness Collaboration (RVASC) [17] (*n* = 1455). Data are shown as mean ± standard deviation.

**Table 1 genes-11-00055-t001:** Analysis of Ehlers–Danlos syndromes by type

	Classical (*n* = 10)	Hypermobile (*n* = 13)	Vascular (*n* = 8)	Other/Unspecified (*n* = 29)	*P* Value	All Patients (*n* = 60)
Age (years)	42 (15–63)	34 (13–51)	38 (13–70)	34 (13–66)	0.557	40 (13–70)
Sex (M/F)	0/9	2/11	2/7	7/22	-	11/49
Height (m)	1.63 ± 0.04	1.67 ± 0.05	1.62 ± 0.12	1.66 ± 0.08	0.336	1.65 ± 0.08
Weight (kg)	74.6 ± 23.5	68.8 ± 12.6	65.3 ± 16.4	68.0 ± 21.3	0.804	70.1 ± 21.8
BMI (kg/m^2^)	27.9 ± 8.0	24.5 ± 4.0	24.5 ± 4.1	24.7 ± 7.7	0.593	25.1 ± 6.6
Supine Hemodynamics						
Systolic BP (mmHg)	116 ± 11	121 ± 7	117 ± 11	120 ± 13	0.655	119 ± 11
Diastolic BP (mmHg)	68 ± 8	68 ± 10	62 ± 11	67 ± 7	0.356	66 ± 8
HR (beats/min)	77 ± 10	77 ± 10	66 ± 8	74 ± 14	0.200	74 ± 13
Central PWV (m/s)	4.91 ± 0.56	4.82 ± 0.38	4.89 ± 0.47	4.59 ± 0.20	0.873	4.73 ± 0.16
Peripheral PWV (m/s)	7.45 ± 0.39	7.24 ± 0.33	7.39± 0.29	7.12 ± 0.17	0.810	7.23 ± 1.00
Standing Hemodynamics						
Systolic BP (mmHg)	110 ± 19	118 ± 21	93 ± 24	121 ± 14	0.003	115 ± 20
Diastolic BP (mmHg)	75 ± 11	81 ± 12	69 ± 14	73 ± 10	0.087	75 ± 12
HR (beats/min)	91 ± 14	92 ± 12	85 ± 17	89 ± 16	0.719	90 ± 15

Body mass index (BMI), blood pressure (BP), mean arterial pressure (MAP), heart rate (HR), pulse wave velocity (PWV). Data are shown as mean (min-max) or mean ± standard deviation.

**Table 2 genes-11-00055-t002:** Correlations of pulse wave velocity to orthostatic hemodynamics in Ehlers-Danlos syndromes

	Carotid to Femoral PWV	Carotid to Radial PWV
SBP (supine)	0.387 *	0.076
SBP (seated)	0.399 *	0.098
SBP (standing)	0.199	0.008
Δ SBP (standing-seated)	−0.077	−0.066
DBP (supine)	0.400 *	0.322 *
DBP (seated)	0.204	0.383 *
DBP (standing)	0.078	0.323 *
Δ DBP (standing-seated)	−0.158	−0.062
HR (supine)	0.015	0.185
HR (seated)	0.044	0.234
HR (standing)	−0.039	0.165
Δ HR (standing-seated)	−0.111	−0.048

Pearson’s r-correlations are shown. Change in (Δ), Systolic blood pressure (SBP), diastolic blood pressure (DBP), heart rate (HR). * Significant correlation at *p* ≤ 0.05 level.

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
