# Peer review of "Arterial Elasticity in Ehlers-Danlos Syndromes"

_genes, 2020, doi:10.3390/genes11010055_

Round 1

Reviewer 1 Report

This is a well conceived and well-written manuscript describing measurements of arterial elasticity in those with EDS. The data are presented in a clear and informative manner. The novel findings in the paper are that those with a range of types of EDS have increased arterial elasticity as measured by lower pulse wave velocity (PWV) compared to published normal values. This paper adds an important new physiologic finding to the literature on orthostatic intolerance in EDS, and has the advantage of being larger than most previous studies on the topic.

I think the authors should acknowledge three additional limitations in the study in section 4.6.

a. The first is that the study did not involve contemporaneous healthy controls, which raises questions about the reliability and comparability of PWV measurement techniques between the NIA study data and the published norms. Did the published norms use the older PWV methods (piezo crystal microphones) or the same ultra-fast ultrasound technique used in this study? If available, perhaps the authors could add something about the comparability of PWV techniques, inter-observer measurement differences, etc. The differences in Figure 1 comparing the NIA EDS patients and the published norms by age are substantial, and would clearly still be valid after accounting for methodological differences in PWV measurement, so acknowledging this limitation does not detract from the main findings.

b. The second is that the numbers of individuals with confirmed classical or hypermobile EDS were relatively small, and these interesting data will need to be replicated in larger, more homogeneous EDS subgroup samples than were available to them from the NIA cohort.

c. Given the new 2017 EDS definitions, it will be important to determine in subsequent studies whether patients with various EDS sub-types differ from those with hypermobility spectrum disorder as defined by the 2017 criteria (some of the 29 Other/Unspecified patients in the NIA study might not meet current EDS consensus definitions).  

Minor points:

1. There is a typo in the Abstract, which on line 19 lists the sample as containing 49 males instead of 49 females.

2. On line 49, reference 3 and 6 should be listed, not just reference 3. 

Author Response

This is a well conceived and well-written manuscript describing measurements of arterial elasticity in those with EDS. The data are presented in a clear and informative manner. The novel findings in the paper are that those with a range of types of EDS have increased arterial elasticity as measured by lower pulse wave velocity (PWV) compared to published normal values. This paper adds an important new physiologic finding to the literature on orthostatic

I think the authors should acknowledge three additional limitations in the study in section 4.6.

The first is that the study did not involve contemporaneous healthy controls, which raises questions about the reliability and comparability of PWV measurement techniques between the NIA study data and the published norms. Did the published norms use the older PWV methods (piezo crystal microphones) or the same ultra-fast ultrasound technique used in this study? If available, perhaps the authors could add something about the comparability of PWV techniques, inter-observer measurement differences, etc. The differences in Figure 1 comparing the NIA EDS patients and the published norms by age are substantial, and would clearly still be valid after accounting for methodological differences in PWV measurement, so acknowledging this limitation does not detract from the main findings.

            Thank you for asking us to clarify these important points. We have added a lack of a contemporaneous control group to the limitations. The NIA study and published norms used the tonometry probe technique for measurement of pulse wave velocity measurement.

The second is that the numbers of individuals with confirmed classical or hypermobile EDS were relatively small, and these interesting data will need to be replicated in larger, more homogeneous EDS subgroup samples than were available to them from the NIA cohort.

            We acknowledge the diversity of cohort being a potential problem. However, it is very difficult to confirm hypermobile EDS due to lack of genetic markers. In the future, we would like to study pulse wave velocity in larger cohorts of people with hypermobile and classical EDS.

Given the new 2017 EDS definitions, it will be important to determine in subsequent studies whether patients with various EDS sub-types differ from those with hypermobility spectrum disorder as defined by the 2017 criteria (some of the 29 Other/Unspecified patients in the NIA study might not meet current EDS consensus definitions).  

            These data were collected prior to 2017. It is possible that some participants that were classified as hypermobile or unspecified EDS would instead be classified as having hypermobility spectrum disorder. We have noted this potential situation in the limitations section (4.6). It will be interesting to investigate whether pulse wave velocity is affected in hypermobility spectrum disorders as well or if this is specific to EDS.

Minor points:

There is a typo in the Abstract, which on line 19 lists the sample as containing 49 males instead of 49 females.

Thank you for bringing this to our attention. We have revised the abstract.

On line 49, reference 3 and 6 should be listed, not just reference 3. 

Reference 6 has been added to line 49 as you suggested.

Reviewer 2 Report

This is a work evaluating arterial stiffness in people with EDS related to Orthostatic intollerance.

I think that there are many points to clarify:

First of all authors wrote about 60 patients with EDS with the following distribution in the table in subgroups:  Classical
(n = 10), Hypermobile (n =13), Vascular (n = 8) and Other/Unspecified (n = 29). I think that is not acceptable considerate so many people in the Unspecified Group of EDS patients; please indicate what "unspecified" means and why this Group is so big.

Please indicate in methods section reference from which you have take the Pulse Wave Velocity value of healthy Group's .

If current diagnostic criteria for orthostatic intolerance involves hemodynamic measurements from the supine to standing posture after at least 10 minutes, which is the reason for your choise to  measure  Orthostatic vital signs following 5 minutes in the supine posture then after 5 minutes sitting then 5 minutes of standing?

Please update references with more and new papers.

Author Response

This is a work evaluating arterial stiffness in people with EDS related to Orthostatic intollerance.

I think that there are many points to clarify:

First of all authors wrote about 60 patients with EDS with the following distribution in the table in subgroups:  Classical
(n = 10), Hypermobile (n =13), Vascular (n = 8) and Other/Unspecified (n = 29). I think that is not acceptable considerate so many people in the Unspecified Group of EDS patients; please indicate what "unspecified" means and why this Group is so big.

            We recognize the limitation of having a large unspecified EDS group and we have further emphasized this in section 4.6. It is important to note that these data were collected prior to the 2017 reclassification of EDS and it was more difficult to confirm hypermobile EDS prior to the change in diagnostic guidelines. Most of the unspecified group would likely be classified as hypermobile EDS or hypermobility spectrum disorders based on current diagnostic criteria.

Please indicate in methods section reference from which you have take the Pulse Wave Velocity value of healthy Group's .

            Thank you for the suggestion. We have described the methods used to collect the reference values in healthy subjects in the methods.

If current diagnostic criteria for orthostatic intolerance involves hemodynamic measurements from the supine to standing posture after at least 10 minutes, which is the reason for your choise to  measure  Orthostatic vital signs following 5 minutes in the supine posture then after 5 minutes sitting then 5 minutes of standing?

            These data were collected as part of a larger study on heredity disorders of connective tissue and a full assessment of orthostatic intolerance was not one of the original study goals; we did not choose for orthostatic vitals to be collected this way. In addition, participants remained on any prescribed medications during the study which could affect orthostatic vital signs. We have noted these limitations in section 4.6.

Please update references with more and new papers.

            We have referenced several manuscripts that were published after 2015. Unfortunately, there are not many studies on pulse wave velocity in EDS to reference.

Round 2

Reviewer 2 Report

Regarding the first point you have answered that data were collected prior to the 2017 reclassification of EDS.

I want accept this explanation but I still consider so big the group of "unclassified" patients and this is not so justified.

Regarding the references I suggest to add at least these articles concerning the cardiovascular autonomic profile in JHS/EDS-HT patients i.e:

Biomed Res Int. 2017;2017:9161865. Orthostatic Intolerance and Postural Orthostatic Tachycardia Syndrome .....

J Neurol Sci. 2014 May 15;340(1-2):99-102 Ehlers-Danlos Syndrome and Postural Tachycardia Syndrome: a relationship study.

Author Response

Regarding the first point you have answered that data were collected prior to the 2017 reclassification of EDS.

I want accept this explanation but I still consider so big the group of "unclassified" patients and this is not so justified.

We recognize that the unclassified group is large and is a limitation to the study. We did not sufficiently describe how participants were stratified and have expanded on this in the methods section. Briefly, several patients with EDS have symptoms reflected of multiple types of EDS, especially classical and hypermobile. Prior to the 2017 reclassification and subsequent hypermobile EDS diagnostic checklist, some patients with hypermobile EDS were labeled "classical-like."  It is very difficult to be certain of hypermobile EDS since there is no known genetic marker. In this study, if the specific type of EDS was uncertain participants were categorized as unclassified EDS. We hope this addresses your concerns.

Regarding the references I suggest to add at least these articles concerning the cardiovascular autonomic profile in JHS/EDS-HT patients i.e:

Biomed Res Int. 2017;2017:9161865. Orthostatic Intolerance and Postural Orthostatic Tachycardia Syndrome .....

J Neurol Sci. 2014 May 15;340(1-2):99-102 Ehlers-Danlos Syndrome and Postural Tachycardia Syndrome: a relationship study.

We have added the references you suggested.